# Collaborative Refining for Learning from Inaccurate Labels

**Bin Han, Yi-Xuan Sun, Ya-Lin Zhang, Libang Zhang, Haoran Hu**
**Longfei Li, Jun Zhou$^{†}$, Guo Ye, Huimei He**
Ant Group
{binlin.hb, xuan.syx, lyn.zyl, libang.zlb, hhr327996,
longyao.llf, jun.zhoujun, yeguo.yg, huimei.hhm}@antgroup.com

## Abstract

This paper considers the problem of learning from multiple sets of inaccurate labels, which can be easily obtained from low-cost annotators, such as rule-based annotators. Previous works typically concentrate on aggregating information from all the annotators, overlooking the significance of data *refinement*. This paper presents a collaborative refining approach for learning from inaccurate labels. To refine the data, we introduce the *annotator agreement* as an instrument, which refers to whether multiple annotators agree or disagree on the labels for a given sample. For samples where some annotators *disagree*, a comparative strategy is proposed to filter noise. Through theoretical analysis, the correlations among multiple sets of labels, the respective models trained on them, and the true labels are uncovered, so that relatively reliable labels can be identified. For samples where all annotators *agree*, an aggregating strategy is designed to mitigate potential noise. Guided by theoretical bounds on loss values, a sample selection criterion is introduced and improved to be more robust against potentially problematic values. Through these two modules, all the samples are refined during training, and these refined samples are used to train a lightweight model simultaneously. Extensive experiments are conducted on benchmark and real-world datasets, which demonstrate the superiority of the proposed framework.

## 1 Introduction

Deep learning has recently demonstrated remarkable achievements across a wide array of applications [5, 6, 9, 15, 33]. The cornerstone of its success rests on the availability of high-quality datasets. However, in industrial environments, obtaining accurate labels can often be costly and time-consuming. For example, in financial scenarios, it is challenging for human annotators to accurately assign labels to predict whether a user will default. Currently, many banks and companies utilize rules as low-cost autonomous annotators, especially in the early stages of some financial products. In healthcare scenarios, many rule-based algorithms are also employed to identify patients at risk of developing certain conditions, allowing for early intervention and potentially preventing more severe complications. Although these annotators can provide labels in a time- and cost-efficient manner [7, 44, 45], a single low-cost annotator usually produces biased or inaccurate labels in practice [8, 13, 18, 24, 29, 40].

On the other hand, multiple annotators can provide diverse perspectives with different insights and knowledge to mitigate errors and biases of individual annotators [30, 45]. Learning from multiple sets of inaccurate labels provided by multiple annotators has garnered widespread attention and

---

$^{†}$Corresponding author.

38th Conference on Neural Information Processing Systems (NeurIPS 2024).

relative works can be divided into two main categories. One stream of research focuses on advanced aggregation algorithms that infer the true labels before or during the training stage. The simplest aggregation algorithm is Majority Voting [38], which treats the labels equally by voting, while the approach known as Weighted Majority Voting [20] uses a weight vector to model annotators' expertise. Enhanced Bayesian Classifier Combination (EBCC) [22] tends to infer truth by modeling the correlation between annotators.

Others focus on training models under the supervision of all annotations. By viewing the ground-truth labels as latent variables, some methods that consider the relationships among multiple sets of labels have been proposed to infer true labels based on the Expectation-Maximization (EM) algorithm [1, 2, 34, 43]. Despite the effectiveness of these EM algorithms, they suffer from computational complexity during the training phase. In recent studies, an increasing number of studies focus on end-to-end learning. These end-to-end algorithms [3, 7, 10, 21, 27, 45, 30] can directly learn from multiple sets of noisy labels and map these noisy labels to part of the model (e.g., transition matrix) , encouraging the model to learn knowledge from all noisy labels collectively [45].

Previous works typically concentrate on aggregating information from all the annotators through label aggregation techniques (e.g., voting) or using labels from all annotators in an end-to-end manner (e.g., mapping multiple sets of noisy labels to part of the model). Most related works often neglect the significance of data *refinement*. Although some methods have recognized the need for data refinement [21, 31, 42], they merely adopt the small-loss criterion [11] to filter noisy samples, and refinement is not their core contribution or their key point. On the contrary, we contend that refining a relatively clean dataset during training is the central point, which can alleviate the demands placed on model design and enhance the model's performance. To refine the dataset, an essential step is to assess the reliability of labels from multiple sets. We consider *annotator agreement* as an instrument, which refers to whether multiple annotators agree or disagree on the labels for a given sample. In this paper, we leverage the annotator agreement information and propose a novel framework named **C**ollaborative **R**efining for **L**earning from inaccurate labels (CRL). For samples where some annotators disagree, labels are ambiguous, and learning from all sets of labels can degrade the model's performance, a comparative strategy is proposed to mitigate noise. For samples where all annotators agree, an aggregating strategy is designed to filter out potential noise. The main contributions of this work can be described as follows:

- For samples where some annotators disagree, we conduct theoretical analysis to uncover the correlations among multiple sets of labels, the respective models trained on them, and the true labels. Guided by theoretical insights, a method called **L**abel **R**efining for samples with **D**isagreements (LRD) is proposed to identify the most reliable label by comparing loss values.

- For samples where all annotators agree, we analyze the theoretical bounds on loss values. Based on these bounds, a method called **R**obust **U**nion **S**election (RUS) is proposed, in which we propose a loss-based selection criterion to select trustworthy samples and improve it to be more robust against potentially problematic values.

- Comprehensive experiments are conducted on benchmark and real-world datasets, which demonstrate the effectiveness of our framework. Moreover, our framework is designed to be independent of any specific model architecture, making it compatible with most existing methods, which is confirmed by further experiments.

## 2   Preliminaries

This paper concentrates on binary classification problems with multiple sets of inaccurate labels. Let $D = \{\boldsymbol{x}_i, \tilde{\boldsymbol{y}}_i\}_{i=1}^N$ be a dataset consisting of $N$ instances labeled by $R$ annotators. $\boldsymbol{x}_i \in \mathcal{X} \subseteq \mathbb{R}^d$ is the feature vector of the $i$-th instance and $\tilde{\boldsymbol{y}}_i = \{\tilde{y}_i^r\}_{r=1}^R$ is a $R$-dimensional vector representing the labels provided by $R$ annotators, $\tilde{y}_i^r \in \{0, 1\}$. Denote $y_i^*$ as the unobserved ground-truth label for the $i$-th instance. $y_i^*$ is considered as a latent variable decided by a latent function $f^*$, *i.e.*, $f^*(\boldsymbol{x}_i) = y_i^*$. $f_\Theta$ denotes a classifier parameterized by $\Theta$, and $f_\Theta(\boldsymbol{x}_i) = \hat{p}(\boldsymbol{x}_i)$ denotes the predicted probability after the activation function, i.e., sigmoid. The binary cross-entropy loss function of the pair $(\boldsymbol{x}_i, \tilde{y}_i^r)$ and model $f_\Theta$ is

$$\ell(f_\Theta(\boldsymbol{x}_i), \tilde{y}_i^r) = \tilde{y}_i^r \log(f_\Theta(\boldsymbol{x}_i)) + (1 - \tilde{y}_i^r) \log(1 - f_\Theta(\boldsymbol{x}_i)). \tag{1}$$

The goal of learning from multiple annotators is to get the optimal classifier $f_{\Theta^*}$ that satisfies $f_{\Theta^*}(\boldsymbol{x}_i) = f^*(\boldsymbol{x}_i)$.

For the model architecture, several shared embedding layers are used to jointly extract information, followed by $R + 1$ submodels $\{f_{\Theta_r}\}_{r=1}^{R+1}$: $R$ submodels $\{f_{\Theta_r}\}_{r=1}^{R}$ are responsible for making predictions based on $R$ sets of labels, and the $(R + 1)$-th submodel $f_{\Theta_{R+1}}$ generates the final prediction. These $R + 1$ submodels are simultaneously trained, with $\{f_{\Theta_r}\}_{r=1}^{R}$ learning from $R$ sets of labels while $f_{\Theta_{R+1}}$ concurrently training from the refined data.

# 3 Method

In this paper, we present a novel framework called collaborative refining for learning from inaccurate labels (CRL). Within the framework, *annotator agreement* is utilized to partition the samples into two categories, i.e., samples where annotators disagree or agree. This concept facilitates the development of targeted strategies for each category.

## 3.1 Collaborative Refining Framework

Based on the *annotator agreement*, the whole dataset can be partitioned into two parts: $D_d$ which contains samples where some annotators disagree, i.e., $\exists r_0, r_1 \subseteq \{1, ..., R\}, \tilde{y}_i^{r_0} \neq \tilde{y}_i^{r_1}$; $D_a$ which contains samples where all annotators agree, i.e., $\forall r_0, r_1 \subseteq \{1, ..., R\}, \tilde{y}_i^{r_0} = \tilde{y}_i^{r_1}$. As mentioned in the introduction, for $D_d$, the labels are ambiguous, and learning from all sets of labels may degrade the model's performance. To this end, we propose Label Refining for samples with Disagreements (LRD) to select relatively reliable labels. For $D_a$, where labels are more reliable but not immune to errors. To address this, we present Robust Union Selection (RUS) to select trustworthy samples.

In the framework, through LRD and RUS, $R$ submodels $\{f_{\Theta_r}\}_{r=1}^{R}$ collaboratively refine unreliable dataset $D_d$ and $D_a$ into relatively reliable dataset $D_d^*$ and $D_a^*$. $D_d^*$ and $D_a^*$ are utilized to train the final submodel $f_{\Theta_{R+1}}$ with binary cross-entropy loss function as shown in Eq. (1). For prediction, $f_{\Theta_{R+1}}$ is utilized. Note that RUS and LRD operate concurrently and share an identical set of $R + 1$ submodels. Our framework improves model performance exclusively by refining information, which is independent of any particular backbone, allowing for seamless substitution and flexibility in the model structure. The complete CRL framework is shown in Algorithm 1.

## 3.2 Label Refining for Samples with Disagreements

When some annotators disagree, learning from all sets of labels may degrade the model's performance. A rational idea is to figure out which label is more reliable. Through theoretical analysis, we uncover the relationships among multiple sets of labels, the ground-truth label, and the submodels' predictions for a given sample, which in turn guide the design of the algorithm.

We follow the widely used class-conditional noise assumption [10, 26], i.e., $p(\tilde{y}^r | y^*, \boldsymbol{x}) = p(\tilde{y}^r | y^*), \forall r \in \{1, \ldots, R\}$. Under this assumption, the noise transition matrix for the $r$-th annotator can be formulated as $T^r \in \mathbb{R}^{2 \times 2}$, where $T_{ij}^r = p(\tilde{y}^r = j | y^* = i)$ denotes the probability of an $i$-th class sample flipped into the $j$-th class for the $r$-th annotator.

Inspired by Gui et al. [11], we give the following theorem about the relationships among multiple sets of labels, the ground-truth label, and the submodels' predictions, proof can be found in Appendix A.1. Assume neural networks $f_{\Theta_0}, f_{\Theta_1}$ are used to minimize the expected loss using labels from two annotators respectively.

**Theorem 1.** *Let $(\boldsymbol{x}, y^*, \tilde{y}^0, \tilde{y}^1)$ be any sample with ground-truth label $y^*$ and two conflicting labels $\tilde{y}^0$, $\tilde{y}^1$ from two annotators, i.e., $\tilde{y}^0 \neq \tilde{y}^1$. Assume $T^0$ and $T^1$ satisfy $T_{ii}^0 > 0.5$ and $T_{ii}^1 > 0.5$, $\forall i \in \{0, 1\}$, $\ell(f_{\Theta_0^*}(\boldsymbol{x}), \tilde{y}^0) < \ell(f_{\Theta_1^*}(\boldsymbol{x}), \tilde{y}^1)$ if and only if $y^* = \tilde{y}^0$.*

**Remark.** Theorem 1 indicates that when the diagonal elements of two noise transition matrices are greater than 0.5, if two models are trained with these two sets of labels, for a sample $\boldsymbol{x}$ on which these two annotators disagree, the more reliable label can be selected by comparing the loss values in their respective models.

Then, the above theorem from the scenario of two annotators can be expanded into multiple annotators, proof can be found in Appendix A.2. Assume a series of neural networks $\{f_{\Theta_r}\}_{r=1}^{R}$ are used to minimize the expected loss using labels from $R$ annotators respectively.

**Corollary 1.** *Let $(\boldsymbol{x}, y^*, \{\tilde{y}^r\}_{r=1}^R)$ be any sample with ground-truth label $y^*$ and $R$ conflicting labels $\{\tilde{y}^r\}_{r=1}^R$ from $R$ annotators, i.e., $\exists r_0, r_1 \subseteq \{1, ..., R\}$, $\tilde{y}^{r_0} \neq \tilde{y}^{r_1}$. Assume $T_{ii}^r > 0.5$, $\forall i \in \{0, 1\}$ and $r \in \{1, ..., R\}$, if $\ell(f_{\Theta_k^*}(\boldsymbol{x}), \tilde{y}^k) = \min(\{\ell(f_{\Theta_r^*}(\boldsymbol{x}), \tilde{y}^r)\}_{r=1}^R)$, $y^* = \tilde{y}^k$.*

**Remark.** Corollary 1 indicates that if $R$ models are trained with $R$ sets of labels, when the observed labels of sample $\boldsymbol{x}$ are not the same, we can get the most reliable label for this sample by choosing the label which has the smallest loss value.

Corollary 1 encourages us to infer the most reliable labels of the samples where some annotators disagree. Based on Theorem 1 and Corollary 1, we propose our LRD method to deal with $D_d$. As mentioned in the preliminaries, $R$ single-label submodels are trained. The training loss for these $R$ submodels can be written as:

$$\mathcal{L}_{BCE} = \frac{1}{NR} \sum_{i=0}^N \sum_{r=1}^R w^r \tilde{y}_i^r \log(f_{\Theta_r}(\boldsymbol{x}_i)) + (1 - \tilde{y}_i^r) \log(1 - f_{\Theta_r}(\boldsymbol{x}_i)), \tag{2}$$

where $w^r$ is a weighting factor introduced to address the predictive bias for positive and negative samples which may arise due to sample imbalance. Denote the number of positive labels in $r$-th set of labels as $n_1^r$ and that of negative labels as $n_0^r$, a common strategy for setting $w^r$ is $w^r = \frac{n_0^r}{n_1^r}$.

Through these submodels, we identify the most reliable labels by comparing their respective loss values over the same sample. Based on Corollary 1, for sample $(\boldsymbol{x}_i, \{\tilde{y}_i^r\}_{r=1}^R) \in D_d$, we get the refined label $\tilde{y}_i^*$ by

$$\tilde{y}_i^* = \tilde{y}_i^k, \tag{3}$$

where the most reliable index $k$ for sample $\boldsymbol{x}_i$ is acquired through:

$$k = \underset{r}{\operatorname{argmin}} \{\ell(f_{\Theta_r}(\boldsymbol{x}_i), \tilde{y}_i^r)\}_{r=1}^R. \tag{4}$$

In this way, for any instance $(\boldsymbol{x}_i, \{\tilde{y}_i^r\}_{r=1}^R)$, we can refine it into $(\boldsymbol{x}_i, \tilde{y}_i^*)$. Then we can construct the refined dataset $D_d^*$ and utilize it to train the submodel $f_{\Theta_{R+1}}$ with the binary cross-entropy loss. This submodel can be deployed and utilized for final prediction. In practice, these refined labels are held constant after several training epochs to mitigate the over-fitting issue. The procedure of LRD is included in Algorithm 1. Note that LRD and RUS operate concurrently, with LRD focusing on $D_d$ and RUS on $D_a$.

### 3.3 Robust Union Selection

For samples where all annotators agree, the labels are generally presumed to be reliable, however, they are not immune to errors. For instance, if the annotators have similar defects, although the annotators agree, the labels may still be inaccurate. To this end, guided by theoretical bounds on loss values, we introduce a loss-based selection criterion and modify it to be more robust against potentially problematic values.

Inspired by the small-loss selection criterion mentioned in many works under single noise label scenario [11, 13, 23], small-loss data tends to be more clean. However, relying solely on the prediction provided by one of the submodels is unstable. Once the selection is wrong, the inferiority of accumulated errors will arise [41]. Since we have $R$ predictions for each sample with $R$ sets of labels from our model, naturally, average loss values can be utilized, which can be denoted as:

$$\tilde{\mu}_i = \frac{1}{R} \sum_{r=1}^R \ell(f_{\Theta_r}(\boldsymbol{x}_i), \tilde{y}_i^r), \tag{5}$$

which is more stable and serves as an estimation of the mean $\mu$. Thanks to the $R$ sets of labels provided by $R$ annotators, these submodels can be diverse, which can ease the inferiority of accumulated errors [13].

However, simply taking the average can also lose some correctly labeled samples. This issue originates from two aspects: *annotator defects* and *hard samples*. Firstly, different annotators may differ in quality or be suitable for different kinds of samples, which leads to the situation where some submodels may struggle with prediction and incur a larger loss for some correctly labeled samples.

Secondly, some correctly labeled samples may be difficult to learn, as a result, submodels may give unstable predictions to them. Note that this issue is avoided in LRD since we use the minimum loss value of the same sample across submodels there. To address this issue, a robust criterion for sample selection is defined.

We begin by defining a non-decreasing smooth function $\phi$ for loss values, which can be written as:

$$\phi(z) = \log(1 + z + \frac{z^2}{2}), \tag{6}$$

where $z$ is a positive variable, this function is inspired by the Taylor expansion of the exponential function. Loss values can be easily substituted into this equation to serve as an *soft estimation* of the underlying mean $\mu$. Eq. (6) can reduce the side effect of extremum, which can mitigate *annotator defects* and *hard samples* issues. Based on Eq. (6), the robust average loss $\tilde{\mu}_i^\phi$ can be written as:

$$\tilde{\mu}_i^\phi = \frac{1}{R} \sum_{r=1}^{R} \phi(\ell(f_{\Theta_r}(\boldsymbol{x}_i), \tilde{y}_i^r)). \tag{7}$$

To enhance the robustness of $\tilde{\mu}_i^\phi$, the predictions from historical models are introduced. Denote the submodel's parameters $\Theta_r$ in the $t$-th epoch as $\Theta_r^t$, the chosen set of epochs as $T$, then Eq. (7) can be rewritten as follows:

$$\tilde{\mu}_i^\phi = \frac{1}{R|T|} \sum_{r=1}^{R} \sum_{t \in T} \phi(\ell(f_{\Theta_r^t}(\boldsymbol{x}_i), \tilde{y}_i^r)), \tag{8}$$

where $|T|$ represents the total number of the selected epochs, for instance, we can choose the latest five epochs or some fixed epochs as $T$ during the training process.

Based on the soft estimation $\tilde{\mu}_i^\phi$ of the underlying mean $\mu$, we further introduce the lower bound of the underlying mean $\mu$ to give correctly labeled samples which are discarded due to *annotator defects* or because they are *hard samples* a chance to be selected. Inspired by Xia et al. [39], we give the following theorem about the lower bound. The proof is similar to Theorem 1 in [39].

**Theorem 2.** *Let $\{z_j\}_{j=1}^n$ be an observation set with mean $\mu$ and variance $\sigma^2$. By utilize a non-decreasing function $\phi(z) = \log(1 + z + \frac{z^2}{2})$, we have*

$$\mu \geq \frac{1}{n} \sum_{j=1}^{n} \phi(z_j) - \frac{\sigma^2(n + \frac{\sigma^2 \log(2n)}{n^2})}{n - \sigma^2}, \tag{9}$$

*with probability at least $1 - \frac{1}{n}$.*

**Remark.** This theorem defines the lower bound of the mean of an observation set. The first term in the lower bound is a robust average, which can reduce the side effects of extremum. The second term is related to the number of observed values and the variance.

Based on Theorem 2, our selection criterion $c_i$ for $\boldsymbol{x}_i$ is defined as:

$$c_i = \tilde{\mu}_i^\phi - \frac{\tilde{\sigma}_i^2(n + \frac{\tilde{\sigma}_i^2 \log(2n)}{n^2})}{n - \tilde{\sigma}_i^2} \tag{10}$$

where $n$ is the number of observed loss values for $\boldsymbol{x}_i$, i.e., $n = R|T|$ as shown in Eq. (8), $\tilde{\sigma}_i^2$ is the variance of observed loss values for $\boldsymbol{x}_i$. Since we don't know the distribution of the loss values for $\boldsymbol{x}_i$ in submodels, $\tilde{\sigma}_i^2$ serves as an approximation of the latent true variance.

Generally speaking, the second term in Eq. (10) gives samples with relatively large variance a chance to be selected for training. If we use the normal average loss, as shown in Eq. (5), these samples may be discarded due to *annotator defects* or because they are *hard samples*. Thanks to Eq. (10), they can be included in the selection; if these samples consistently yield large loss values throughout the training process and across the majority of submodels, they will be discarded in subsequent epochs.

By using the selection criterion in Eq. (10), we can sort the samples in $D_a$ and select the smallest $p$ proportion to form $D_a^*$, which can be utilized to train $f_{\Theta_{R+1}}$. In practice, positive and negative samples are selected separately. The process of RUS is included in Algorithm 1.

Note that RUS and LRD operate concurrently, and they share an identical set of $R + 1$ submodels which are trained simultaneously. In RUS, the $R$ submodels $\{f_{\Theta_r}\}_{r=1}^R$ collaboratively refine $D_a$, while in LRD, these submodels are used to refine $D_d$. $f_{\Theta_{R+1}}$ learns from the refined samples.

# 4 Experiment

## 4.1 Dataset

**Benchmark datasets.** All the methods are evaluated on 13 benchmark datasets with two kinds of noise, including five NLP datasets named Agnews, 20News, IMDb, Yelp and Amazon, four tabular datasets named Diabetes, Backdoor, Campaign and Waveform, and four image datasets named Celeba, SVHN, Fashion-MNIST (denote as F-MNIST) and CIFAR-10. Note that these datasets only have ground-truth labels, we simulate three annotators per dataset with label quality $k = 0.3$ through the following two methods:

- **Instance-dependent noise.** Inspired by Zhao et al. [45], $k$ proportion of each dataset is used to train three sets of diverse tree-based models to act as rule-based annotators, i.e., Decision Tree, RandomForest, LightGBM. Their predictions are used as multiple sets of labels.

- **Class-dependent noise.** The labels of the positive samples are randomly preserved at proportions of $k$, $k + 0.1$, and $k + 0.2$ respectively, while interchanging the labels for the remaining positive samples with those of negative samples, thus establishing three sets of labels. Note that these three sets of labels are generated independently.

**Real-world datasets.** Experiments are also conducted on two real-world datasets: CIFAR-10N and Sentiment. Both datasets were published on Amazon Mechanical Turk for annotation.

Details of these datasets and labels can be found in Appendix B.

## 4.2 Compared Methods and Implementation Details

We compare our framework with various methods: (1) Single, in which a three-layer MLP is trained directly with one set of labels; (2) NN-Mjv [38], in which a three-layer MLP is trained using the labels aggregated through majority voting; (3) HE_A and HE_M [37], which train several individual models for each set of inaccurate labels and aggregate their outputs for predictions by averaging and maximizing; (4) CL [26], which trains an end-to-end model with parametric source-specific transition matrices; (5) DN [10], which exploits information from multiple sets of labels with different softmax output layers; (6) Label aggregation methods called Enhanced Bayesian Classifier Combination (EBCC) and Independent Bayesian Classifier Combination (IBCC) [22], three-layer MLPs are trained using the labels inferred by EBCC or IBCC; (7) Weakly supervised end-to-end learner WeaSEL [27]; (8) CoNAL [3], which assumes that the annotation noise is attributed to two sources (common noise and individual noise), and combines these two noise by a Bernoulli random variable; (9) SLF [7], which proposes to make the weight vectors and the confusion matrices data-dependent, and comes up with two regularization methods for the confusion matrix to guide the training process; (10) ADMoE [45], which leverages the Mixture of Experts (MoE) architecture to encourage specialized learning from multiple noisy sources. Note that for a fair comparison, we do not use the noisy-label aware gating in ADMoE which inputs labels during both training and testing.

Most of the compared methods are adopted from their respective codebases. The hyperparameters are set according to the recommendations in their papers. The only modification we made to ensure a fair comparison is to use a three-layer MLP with a hidden dimension of 128 as the backbone.

For our method, as mentioned in the preliminaries, we use a three-layer MLP of hidden dimension 128 to act as shared layers, and $R + 1$ three-layer MLPs of hidden dimension 128 to serve as $R + 1$ submodels. The $(R + 1)$-th submodel $f_{\Theta_{R+1}}$ is utilized for final prediction. There is no hyperparameter in LRD. For RUS, we set the proportion of selected samples $p = 0.8$, and take the 5th epoch and the latest epoch during training as the selected epochs in Eq.( 8). In practice, LRD-generated labels are held constant after 5 training epochs to mitigate the over-fitting issue.

For all of the methods, experiments are conducted with 0.001 learning rate, 100 training epochs, and 256 batch size on MLP with hidden dimension 128 for a fair comparison. For benchmark datasets, 70% of each dataset is utilized for training, 5% for validation, and 25% for testing. For real-world datasets, we use the original training and testing datasets. We report the average AUC of the last 10 epochs as results.

## 4.3 Main Results

**Main results.** Comprehensive experiments are conducted on various datasets, including NLP, image, and tabular data, and we compare our framework against a variety of methods. The main results of our framework and all the compared methods on class-dependent noise and instance-dependent noise are summarized in Table 1. As shown, our framework outperforms all the compared methods in most of the datasets, especially in the class-dependent noise setting. Our framework does not incorporate any special designs at the architectural level, it primarily acquires higher-quality data through collaborative refinement, followed by learning via a MLP, which demonstrates the effectiveness of our framework.

Table 1: Main results with AUC as the evaluation metric. The best results are in bold.

| Noise | Dataset | Methods | | | | | | | | | | | | |
|---|---|---|---|---|---|---|---|---|---|---|---|---|---|---|
| | | Single | NN-Mjv | HE_A | HE_M | CL | DN | NN-EBCC | NN-IBCC | WeaSEL | SLF | CoNAL | ADMoE | Ours |
| Class | AgNews | 0.660 | 0.634 | 0.723 | 0.724 | 0.626 | 0.757 | 0.703 | 0.746 | 0.833 | 0.791 | 0.769 | 0.762 | **0.855** |
| | 20News | 0.746 | 0.684 | 0.755 | 0.756 | 0.749 | 0.729 | 0.796 | 0.779 | 0.824 | 0.768 | 0.778 | 0.765 | **0.849** |
| | IMDb | 0.666 | 0.602 | 0.667 | 0.670 | 0.614 | 0.689 | 0.673 | 0.699 | 0.709 | 0.700 | 0.702 | 0.707 | **0.766** |
| | Yelp | 0.725 | 0.713 | 0.783 | 0.785 | 0.779 | 0.779 | 0.782 | 0.786 | 0.805 | 0.807 | 0.769 | 0.799 | **0.867** |
| | Amazon | 0.586 | 0.567 | 0.681 | 0.687 | 0.631 | 0.661 | 0.635 | 0.657 | 0.718 | 0.679 | 0.672 | 0.652 | **0.775** |
| | Diabetes | 0.648 | 0.610 | 0.576 | 0.592 | 0.633 | 0.686 | 0.657 | 0.663 | 0.680 | 0.577 | 0.696 | 0.646 | **0.728** |
| | Backdoor | 0.530 | 0.535 | 0.681 | 0.687 | 0.651 | 0.765 | 0.668 | 0.771 | 0.640 | 0.816 | 0.716 | 0.814 | **0.937** |
| | Campaign | 0.561 | 0.558 | 0.628 | 0.636 | 0.574 | 0.663 | 0.619 | 0.632 | 0.629 | 0.694 | 0.697 | 0.680 | **0.783** |
| | Waveform | 0.772 | 0.744 | 0.660 | 0.663 | 0.792 | 0.770 | 0.788 | 0.802 | **0.840** | 0.807 | 0.818 | 0.823 | 0.840 |
| | Celeba | 0.738 | 0.710 | 0.723 | 0.725 | 0.784 | 0.849 | 0.758 | 0.768 | 0.782 | 0.851 | 0.859 | 0.824 | **0.891** |
| | SVHN | 0.637 | 0.639 | 0.671 | 0.671 | 0.692 | 0.701 | 0.676 | 0.679 | 0.671 | 0.730 | 0.726 | 0.718 | **0.761** |
| | F-MNIST | 0.607 | 0.590 | 0.684 | 0.691 | 0.682 | 0.667 | 0.664 | 0.631 | 0.678 | 0.723 | 0.705 | 0.737 | **0.776** |
| | CIFAR-10 | 0.541 | 0.573 | 0.574 | 0.576 | 0.596 | 0.570 | 0.590 | 0.590 | 0.587 | 0.587 | 0.634 | 0.625 | **0.655** |
| Instance | AgNews | 0.842 | 0.819 | 0.830 | 0.829 | 0.848 | 0.867 | 0.817 | 0.786 | 0.836 | 0.768 | 0.882 | 0.842 | **0.916** |
| | 20News | 0.795 | 0.855 | 0.847 | 0.841 | 0.823 | **0.872** | 0.855 | 0.846 | 0.830 | 0.859 | 0.828 | 0.833 | 0.864 |
| | IMDb | 0.668 | 0.729 | 0.710 | 0.708 | 0.722 | 0.739 | 0.635 | 0.621 | 0.644 | 0.686 | 0.717 | 0.713 | **0.749** |
| | Yelp | 0.773 | 0.782 | 0.812 | 0.809 | 0.790 | 0.870 | 0.784 | 0.778 | 0.812 | 0.838 | 0.832 | 0.826 | **0.901** |
| | Amazon | 0.745 | 0.690 | 0.774 | 0.772 | 0.777 | 0.767 | 0.696 | 0.673 | 0.715 | 0.660 | 0.790 | 0.772 | **0.804** |
| | Diabetes | 0.747 | 0.731 | 0.749 | 0.748 | 0.719 | 0.778 | 0.728 | 0.722 | 0.747 | 0.772 | 0.767 | 0.765 | **0.818** |
| | Backdoor | 0.646 | 0.572 | 0.610 | 0.613 | 0.640 | 0.593 | 0.597 | 0.616 | 0.606 | 0.526 | 0.581 | 0.758 | **0.792** |
| | Campaign | 0.776 | 0.726 | 0.892 | 0.890 | 0.811 | 0.887 | 0.740 | 0.733 | 0.737 | 0.761 | 0.883 | 0.707 | **0.914** |
| | Waveform | 0.865 | 0.834 | 0.923 | 0.921 | 0.935 | 0.949 | 0.853 | 0.837 | 0.873 | 0.910 | 0.959 | 0.937 | **0.964** |
| | Celeba | 0.854 | 0.918 | 0.885 | 0.883 | 0.909 | 0.825 | 0.920 | 0.922 | 0.936 | 0.933 | 0.914 | 0.916 | **0.944** |
| | SVHN | 0.702 | 0.708 | 0.774 | 0.772 | 0.813 | 0.806 | 0.724 | 0.721 | 0.779 | 0.775 | 0.803 | 0.800 | **0.824** |
| | F-MNIST | 0.814 | 0.801 | 0.888 | 0.887 | 0.856 | 0.840 | 0.792 | 0.799 | 0.819 | 0.801 | 0.870 | 0.888 | **0.912** |
| | CIFAR-10 | 0.730 | 0.702 | 0.778 | 0.777 | 0.772 | 0.764 | 0.696 | 0.688 | 0.747 | 0.632 | 0.777 | **0.799** | 0.790 |

**Performance on real-world scenarios.** Experiments are conducted on these two real-world noisy datasets, with both our methods and the compared methods. As summarized in Table 2, our method surpasses the compared methods on the real-world noisy datasets, these additional results further substantiate the reliability and applicability of our approach in handling real-world noisy labels.

**Results under different label quality.** Further experiments are conducted under different label quality ($k = 0.1, 0.15, 0.2, 0.25, 0.3$) on AgNews, IMDb, Yelp, Diabetes, Celeba and F-MINST with class-dependent noise where NLP, image and tabular data are all involved. The results are shown in Figure 1. For clarity, we only include the four competitive compared methods and the Single baseline in the figure. With the increase in label quality, all methods achieve better performance. Note that our framework consistently outperforms the compared methods in most of the experiments.

## 4.4 Discussion

**Ablation study.** Ablation experiments are conducted on datasets with class-dependent noise where NLP, image, and tabular data are all involved. The results are shown in Table 3. In the ablation experiments, NN-Mjv acts as the baseline. Labels are aggregated for $D_d$ / $D_a$ by majority voting instead of LRD / RUS, without changing any other parts of the framework. We also compare the RUS method with the naive selection method (denoted as N-RUS) by replacing our selection criterion in Eq. (10) with the naive one in Eq. (5). As shown in Table 3, with the assistance of LRD and RUS, an overall improvement of 0.171 in AUC is achieved over the baseline. The results indicate that while the naive selection does work, it's not as effective as RUS. Since LRD and RUS are designed for different parts of the dataset respectively, they can be effectively combined to achieve the best results.

Table 2: Real-world results with AUC as the evaluation metric. The best results are in bold.

| Dataset | Methods | | | | | | | | | | | | |
|---|---|---|---|---|---|---|---|---|---|---|---|---|---|
| | Single | NN-Mjv | HE_A | HE_M | CL | DN | NN-EBCC | NN-IBCC | WeaSEL | SLF | CoNAL | ADMoE | Ours |
| Sentiment | 0.712 | 0.727 | 0.744 | 0.730 | 0.724 | 0.732 | 0.728 | 0.736 | 0.730 | 0.686 | 0.741 | 0.722 | **0.753** |
| CIFAR-10N | 0.791 | 0.853 | 0.788 | 0.786 | 0.788 | 0.807 | 0.850 | 0.849 | 0.851 | 0.821 | 0.816 | 0.761 | **0.866** |

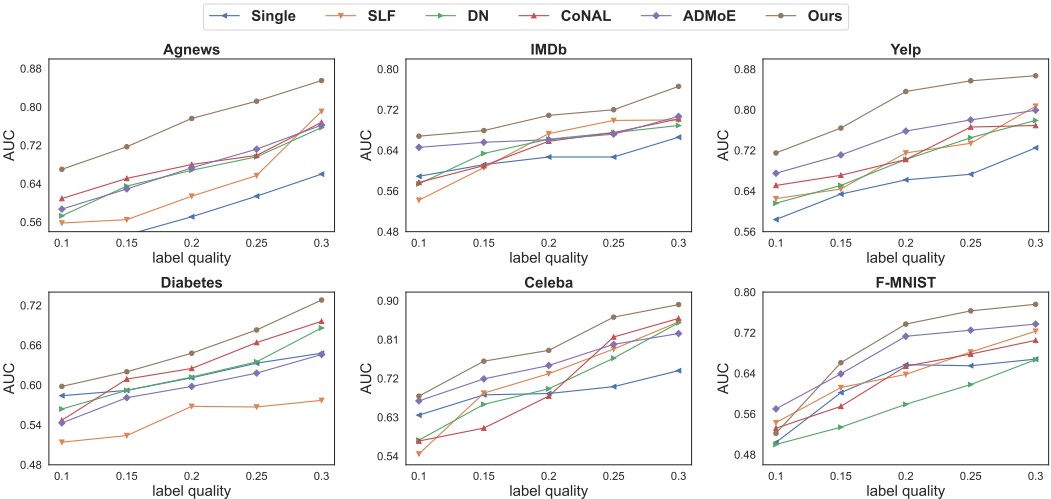

Figure 1: AUC comparison under different label quality $k$.

Through these experiments, we demonstrate the effectiveness of LRD and RUS. These results also indicate that adding LRD to the baseline improves the performance more noticeably than when adding RUS. It is reasonable because LRD is designed to correct labels in $D_d$, while RUS is designed to select samples in $D_a$. As is illustrated before, the samples in $D_d$ are more ambiguous and the samples in $D_a$ are more likely to be correctly labeled, thus greater improvement can be achieved by effectively dealing with $D_d$.

**Quality analysis of the refined data.** Further experiments are conducted to explore how LRD and RUS work. Since the LRD method is designed to aggregate multiple sets of labels into a reliable one, we conduct experiments to show the label quality produced by LRD. We train our model for 5 epochs and collect the labels aggregated by the LRD algorithm on $D_d$. Then we compare the quality of these labels with the three sets of original labels and the labels obtained through voting. The experiments are conducted on thirteen datasets mentioned above with class-dependent noise. The results are shown in Figure 2, the average AUC of these labels is taken as the metric. LRD can consistently provide higher-quality labels. Naturally, our model trained with these labels yields better results than with the original labels.

Since RUS is designed to select samples, experiments are conducted to show the quality of the selected samples. The model is trained for 5 epochs then the selected samples are fetched. RUS is compared with the following two methods: (1) selecting the same proportion of samples randomly; (2) replacing the criterion in RUS with the mean of the loss values in Eq. (5) to select the same proportion of samples. The proportion of selection $p$ is set as $0.8$. Since the total number of ground-truth positive samples stays the same, we compare the number of true positive samples selected by the three methods mentioned above. These experiments are repeated for 100 times and the average improvements over random selection are presented in Figure 3. As shown in Figure 3, on the four datasets including image, NLP, and tabular data, our proposed method has a better performance in selecting samples. Naturally, our model trained with these high-quality samples can get better results.

**Further verification of LRD-generated labels.** The quality of LRD-generated labels is shown in the foregoing discussion in Figure 2, whereby we demonstrate the effectiveness of LRD after training models for five epochs. We further examine LRD's performance in the early training stages. We train

Table 3: Ablation results with AUC as the evaluation metric.

| Methods | Datasets | | | | | | Avg. |
|---|---|---|---|---|---|---|---|
| | AgNews | IMDb | Yelp | Diabetes | Celeba | F-MINST | |
| Baseline | 0.634 | 0.602 | 0.713 | 0.610 | 0.710 | 0.590 | 0.643 |
| +RUS | 0.786 | 0.708 | 0.846 | 0.613 | 0.838 | 0.703 | 0.749 |
| +LRD | 0.832 | 0.733 | 0.859 | 0.668 | 0.840 | 0.735 | 0.778 |
| +LRD and N-RUS | 0.838 | 0.746 | 0.863 | 0.692 | 0.862 | 0.774 | 0.796 |
| +LRD and RUS | **0.855** | **0.766** | **0.867** | **0.728** | **0.891** | **0.776** | **0.814** |

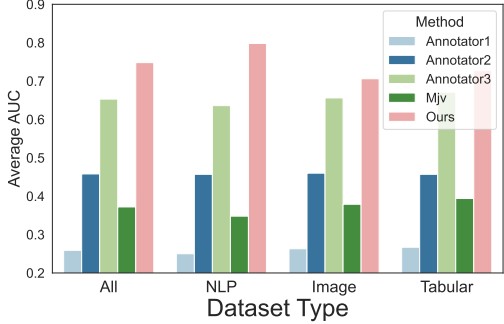

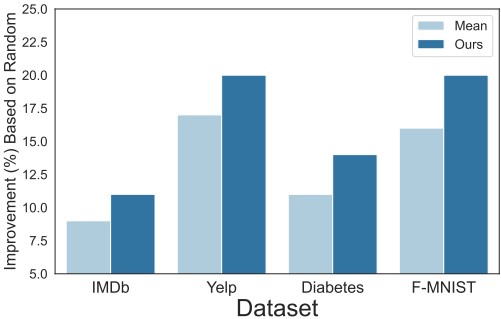

Figure 2: The average AUC on $D_d$ subsets over different types of dataset.

Figure 3: Average improvement(%) in the quantity of true positive samples selected over a random baseline.

our model on datasets with class-dependent noise for 100 / 500 steps and examine the qualities of LRD-generated labels on $D_d$. As shown in Table 4, in the early stages of LRD learning (e.g., 100 steps), the LRD-generated labels already outperform the original labels. With ongoing training, label quality rises (e.g., 500 steps). When LRD-generated labels are frozen at step 100, the final average AUC on these four datasets is 0.835, which is quite close to the average AUC of 0.845 obtained by freezing LRD at step 500. These results indicate that we have considerable flexibility in choosing when to fix the refined labels. In the main results, the LRD-generated labels are frozen after training for five epochs.

**Cooperating with other algorithms.** As mentioned before, our framework tends to collaboratively refine the dataset into a higher-quality one, which is independent of the model architecture. We simply utilize a MLP to learn from the refined dataset, which can be replaced by other algorithms. We train our model for 5 epochs, extract the refined dataset, and complement it with the original three sets of labels to form a new dataset. Then this new dataset is utilized to train SLF, CoNAL, and ADMoE in the same manner as in the main experiments. As shown in Table 5, when cooperating with our framework, ADMoE, SLF, and CoNAL achieve improvement over their original versions, which demonstrates that our framework can cooperate well with the existing methods.

## 5   Conclusion and Future Work

In this paper, we present a framework called Collaborative Refining for Learning from inaccurate labels (CRL), which focuses on learning a model from multiple sets of inaccurate labels. In our framework, we utilize the *annotator agreement* to assess the reliability of labels from multiple sets with disparities and split the dataset into two parts: where some annotators disagree and where all annotators agree. For samples where some annotators disagree, Label Refining for samples with Disagreements (LRD) is proposed to select relatively reliable labels. Based on a theoretical analysis of the relationships among multiple label sets, ground-truth labels, and model predictions, LRD identifies the most reliable label by comparing the loss values. For samples where all annotators agree, Robust Union Selection (RUS) is proposed to select trustworthy samples to form a higher-quality dataset. Guided by theoretical bounds on loss values, RUS introduces a loss-based selection criterion and improves it to be more robust against potentially problematic values. Meanwhile, the

Table 4: Label qualities on $D_d$ and the training results. AUC is the evaluation metric.

| Label sources | AgNews | IMDb | Yelp | Celeba | Avrage |
|---|---|---|---|---|---|
| Best Annotator | 0.647 | 0.630 | 0.671 | 0.656 | 0.651 |
| Voting | 0.340 | 0.348 | 0.418 | 0.366 | 0.368 |
| LRD-100steps | 0.775 | 0.665 | 0.870 | 0.736 | 0.762 |
| LRD-500steps | **0.858** | **0.706** | **0.880** | **0.831** | **0.819** |

Table 5: Cooperation results with AUC as the evaluation metric.

| Methods | Dataset | | | | | | Average |
|---|---|---|---|---|---|---|---|
| | AgNews | IMDb | Yelp | Diabetes | Celeba | F-MINST | |
| ADMoE | 0.762 | 0.707 | 0.799 | 0.646 | 0.824 | 0.737 | 0.746 |
| +Ours | **0.803** | **0.732** | **0.804** | **0.692** | **0.840** | **0.762** | **0.772** |
| Improve(%). | +5.47 | +3.48 | +0.70 | +7.10 | +1.99 | +3.28 | +3.54 |
| SLF | 0.791 | 0.700 | 0.807 | 0.577 | 0.851 | 0.723 | 0.741 |
| +Ours | **0.806** | **0.711** | **0.834** | **0.710** | **0.900** | **0.769** | **0.788** |
| Improve(%). | +1.91 | +1.55 | +3.37 | +23.11 | +5.85 | +6.41 | +6.35 |
| CoNAL | 0.769 | 0.702 | 0.769 | 0.696 | 0.859 | 0.705 | 0.750 |
| +Ours | **0.776** | **0.719** | **0.782** | **0.723** | **0.884** | **0.737** | **0.770** |
| Improve(%). | +0.92 | +2.40 | +1.74 | +3.86 | +2.82 | +4.64 | +2.69 |

refined datasets are used to train a lightweight model that can be deployed and utilized for final prediction. Extensive experiments are conducted to demonstrate the effectiveness of the framework. The framework is designed to be independent of any specific model architecture, making it compatible with most existing methods, which is confirmed by further experiments.

This work currently focuses on the binary classification task with the existence of inaccurate labels, which is fundamental and commonly encountered in practice. This work also lays the foundation for subsequent research of multi-class classification tasks with inaccurate labels. However, in the multi-class scenario, the proposed method needs to be further adapted. For instance, an issue arises with samples where annotators disagree, particularly when all the given labels for a particular sample are incorrect in the multi-class setting. LRD tends to select one of these incorrect labels as the inferred label, negatively impacting the model's performance. In binary classification problems, this situation is naturally avoided because when annotators disagree, one of the labels must be correct. A possible solution could be to develop a more refined measure based on annotator agreement, which could be utilized to decide whether to discard these samples, using LRD or RUS.

In the current approach, the samples are segmented based on the presence or absence of label disagreement, and further processes are performed on the correlated segments. We believe that the sample segmentation strategy can be further improved. To achieve this, we can introduce a new metric: the consistency rate, defined as the proportion of agreement among annotators for the same sample. By establishing a threshold for this consistency rate, we can effectively partition the dataset. Specifically, for samples with a consistency rate exceeding the threshold, RUS is applied, whereas for samples with a consistency rate below the threshold, LRD is utilized. The determination of this threshold is related to the label qualities of the dataset and the number of label sets available, which is suitable for practical applications. Since these ideas are not yet mature enough, we did not include these parts in the current manuscript. We leave these issues for future research.

# Acknowledgments

This work is supported by Ant Group. We thank the anonymous referees for their constructive suggestions and thoughtful comments, which helped to improve the quality of the paper.

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

# Appendix

In the Appendix, we first give the proofs of Theorem 1, Theorem 2, and Corollary 1 in Sections A. The details of datasets and labels are provided in Section B, and the implementation details of the algorithm are provided in Section C. The related works are summarized in Section D.

## A  Proof

### A.1  Theorem 1

*Proof.* Let $\tilde{\boldsymbol{d}} = [\tilde{d}_0, \tilde{d}_1]$ denote the one-hot version of $\tilde{y}$, i.e., $\tilde{d}_{\tilde{y}} = 1$ and $\tilde{d}_i = 0, \forall i \neq \tilde{y}$. For binary cross-entropy loss function $\ell(f_\Theta(\boldsymbol{x}), \tilde{y}) = -\sum_{i=0}^1 \tilde{d}_i \log(\hat{p}_i(\boldsymbol{x}))$, we consider the expected loss on noisy data:

$$
\begin{aligned}
\mathbb{E}_{(\boldsymbol{x}, \tilde{y})}[\ell(f_\Theta(\boldsymbol{x}), \tilde{y})] &= -\mathbb{E}_{(\boldsymbol{x}, \tilde{y})}\Big[\sum_{i=0}^1 \tilde{d}_i \log(\hat{p}_i(\boldsymbol{x}))\Big] \\
&= -\int_{\boldsymbol{x} \in \mathcal{X}} \sum_{j=0}^1 \Big[\sum_{i=0}^1 \tilde{d}_i \log(\hat{p}_i(\boldsymbol{x}))\Big] p(\boldsymbol{x}, \tilde{y} = j) \mathrm{d}\boldsymbol{x} \\
&= -\int_{\boldsymbol{x} \in \mathcal{X}} \Big[\sum_{i=0}^1 \Big[\sum_{j=0}^1 \tilde{d}_i p(\tilde{y} = j|\boldsymbol{x})\Big] \log(\hat{p}_i(\boldsymbol{x}))\Big] p(\boldsymbol{x}) \mathrm{d}\boldsymbol{x} \\
&= -\int_{\boldsymbol{x} \in \mathcal{X}} \Big[\sum_{i=0}^1 \mathbb{E}[\tilde{d}_i|\boldsymbol{x}] \log(\hat{p}_i(\boldsymbol{x}))\Big] p(\boldsymbol{x}) \mathrm{d}\boldsymbol{x}.
\end{aligned}
\tag{11}
$$

Therefore, minimizing the expected loss equals to minimizing $-\sum_{i=0}^1 \mathbb{E}[\tilde{d}_i|\boldsymbol{x}] \log(\hat{p}_i(\boldsymbol{x}))$ for each $\boldsymbol{x} \in \mathcal{X}$. For cross-entropy loss, there are constraints $\sum_{i=0}^1 \hat{p}_i(\boldsymbol{x}) = 1$ and $0 \leq \hat{p}_i(\boldsymbol{x}) \leq 1, \forall i \in \{0, 1\}$. So it can be formalized as the following optimization problem:

$$
\begin{aligned}
&\text{minimize} -\sum_{i=0}^1 \mathbb{E}[\tilde{d}_i|\boldsymbol{x}] \log(\hat{p}_i(\boldsymbol{x})) \\
&s.t. \sum_{i=0}^1 \hat{p}_i(\boldsymbol{x}) = 1, 0 \leq \hat{p}_i(\boldsymbol{x}) \leq 1, \forall i \in \{0, 1\}
\end{aligned}
\tag{12}
$$

Using Lagrange multiplier method, we can derive that $-\sum_{i=0}^1 \mathbb{E}[\tilde{d}_i|\boldsymbol{x}] \log(\hat{p}_i(\boldsymbol{x}))$ is minimized when $\hat{p}_i(\boldsymbol{x}) = \mathbb{E}[\tilde{d}_i|\boldsymbol{x}], \forall i \in \{0, 1\}$. Because $\mathbb{E}[\tilde{d}_i|\boldsymbol{x}] = \sum_{j=0}^1 \mathbb{I}[i = j] p(\tilde{y} = j|\boldsymbol{x}) = p(\tilde{y}^k = i|\boldsymbol{x})$, we have $\hat{p}_i(\boldsymbol{x}) = p(\tilde{y} = i|\boldsymbol{x})$. Then we can obtain

$$
\begin{aligned}
\hat{p}_i(\boldsymbol{x}) = p(\tilde{y} = i|\boldsymbol{x}) &= \sum_{j=0}^1 p(\tilde{y} = i, y = j|\boldsymbol{x}) \\
&= \sum_{j=0}^1 p(y = j|\boldsymbol{x}) p(\tilde{y} = i|y = j, \boldsymbol{x}) \\
&= \sum_{j=0}^1 p(y = j|\boldsymbol{x}) p(\tilde{y} = i|y = j) \\
&= p(\tilde{y} = i|y = y^*) \\
&= T_{y^* i},
\end{aligned}
\tag{13}
$$

where the fourth equation is due to the class-conditional noise assumption and the fifth equation is due to that each $\boldsymbol{x}$ has only one true label $f^*(\boldsymbol{x})$. Therefore, the loss value of sample $(\boldsymbol{x}, \tilde{y})$ is

$$
\ell(f_{\Theta^*}(\boldsymbol{x}), \tilde{y}) = -\log(\hat{p}_{\tilde{y}}(\boldsymbol{x})) = -\log(T_{y^* \tilde{y}}).
\tag{14}
$$

When facing with two conflicting observed labels $\tilde{y}^0$, $\tilde{y}^1$ and two neural network $f_{\Theta_0^*}, f_{\Theta_1^*}$, the losses can be written as

$$\begin{aligned}
\ell(f_{\Theta_0^*}(\boldsymbol{x}), \tilde{y}^0) &= -\log(T_{y^*\tilde{y}^0}^0), \\
\ell(f_{\Theta_1^*}(\boldsymbol{x}), \tilde{y}^1) &= -\log(T_{y^*\tilde{y}^1}^1).
\end{aligned} \tag{15}$$

Then from the relationship between these two losses $\ell(f_{\Theta_0^*}(\boldsymbol{x}), \tilde{y}^0) < \ell(f_{\Theta_1^*}(\boldsymbol{x}), \tilde{y}^1)$, we can get

$$-\log(T_{y^*\tilde{y}^0}^0) < -\log(T_{y^*\tilde{y}^1}^1), \tag{16}$$

which means

$$T_{y^*\tilde{y}^0}^0 > T_{y^*\tilde{y}^1}^1. \tag{17}$$

Because $T^0$, $T^1$ satisfy $T_{ii}^0 > 0.5$, $T_{ii}^1 > 0.5$, $\forall i \in \{0,1\}$, we can get $T_{ij}^0 < 0.5$ and $T_{ij}^0 < 0.5$, $\forall i \in \{0,1\}, i \neq j$. Since $\tilde{y}^0 \neq \tilde{y}^1$, $T_{y^*\tilde{y}^0}^0 > T_{y^*\tilde{y}^1}^1$ equals to $T_{y^*\tilde{y}^0}^0 > T_{y^*(1-\tilde{y}^0)}^1$, only when $y^* = \tilde{y}^0$,

$$T_{y^*\tilde{y}^0}^0 = T_{y^*y^*}^0 > 0.5 > T_{y^*(1-\tilde{y}^0)}^1 = T_{y^*\tilde{y}^1}^1,$$

which satisfy Eq. (17). if $y^* = \tilde{y}^1$,

$$T_{y^*\tilde{y}^0}^0 = T_{y^*(1-y^*)}^0 < 0.5 < T_{y^*y^*}^1 = T_{y^*\tilde{y}^1}^1,$$

which is in conflict with Eq. (17). Theorem 1 is proved.

$\square$

### A.2 Corollary 1

*Proof.* Let $r_0, r_1 \in \{1, ..., R\}$, define $\ell^{r_0}, \ell^{r_1}$ as

$$\begin{aligned}
\ell^{r_0} &= \min(\ell(f_{\Theta_i^*}(\boldsymbol{x}), \tilde{y}^i)), \tilde{y}^i = \tilde{y}^{r_0}, \\
\ell^{r_1} &= \min(\ell(f_{\Theta_i^*}(\boldsymbol{x}), \tilde{y}^i)), \tilde{y}^i = \tilde{y}^{r_1}.
\end{aligned}$$

Then $\ell(f_{\Theta_k^*}(\boldsymbol{x}), \tilde{y}^k) = \min(\{\ell(f_{\Theta_r^*}(\boldsymbol{x}), \tilde{y}^r)\}_{r=1}^R)$ can be written as

$$\ell(f_{\Theta_k^*}(\boldsymbol{x}), \tilde{y}^k) = \min(\ell^{r_0}, \ell^{r_1}). \tag{18}$$

If $y^* = \tilde{y}^{r_0}$, from Theorem 1 and Eq. (14), we can get

$$\ell^{r_0} < -log(0.5) < \ell^{r_1},$$

then $\ell(f_{\Theta_k^*}(\boldsymbol{x}), \tilde{y}^k) = \ell^{r_0}$ which means $\tilde{y}^k = \tilde{y}^{r_0} = y^*$.

Similarly, if $y^* = \tilde{y}^{r_1}$, we can get $\tilde{y}^k = \tilde{y}^{r_1} = y^*$. Corollary 1 is proved.

$\square$

## B    Details of Datasets

As mentioned in Section 4.1, we use thirteen benchmark datasets including NLP, image and tabular datasets, and two real-world datasets.

**Benchmark datasets.** The information of benchmark datasets are summarized in Table 6. Most of them come from [14].

For image datasets, following [14], we use ResNet18[16] which is pretrained on the ImageNet[5] to extract embedding after the last average pooling layer. We utilize the extracted 512-dimensional embeddings as features. We define one of the multi-classes as negative and downsample the remaining classes to 5% of the total instances as positive.

For NLP datasets, we use BERT[6] pretrained on the BookCorpus and English Wikipedia to extract 768-dimensional embeddings as features. As for labels, we define them in the following way: (1) Amazon and Imdb: we regard the original negative class as the anomaly class. (2) 20News and

Table 6: Data description of the thirteen benchmark datasets used in this paper.

| Dataset | Description | | | |
|---|---|---|---|---|
| | # Samples | # Features | # Positives | # Type |
| AgNews | 10000 | 768 | 500 | NLP |
| 20News | 3090 | 768 | 154 | NLP |
| IMDb | 10000 | 768 | 500 | NLP |
| Yelp | 10000 | 768 | 500 | NLP |
| Amazon | 10000 | 768 | 500 | NLP |
| Diabetes | 10927 | 21 | 676 | Tabular |
| Backdoor | 9555 | 196 | 224 | Tabular |
| Campaign | 9785 | 62 | 586 | Tabular |
| Waveform | 3443 | 21 | 100 | Tabular |
| Celeba | 25906 | 39 | 1118 | Image |
| SVHN | 5208 | 512 | 260 | Image |
| F-MNIST | 6315 | 512 | 315 | Image |
| CIFAR-10 | 5263 | 512 | 263 | Image |

AgNews, we set one of the classes as normal and downsample the remaining classes to 5% of the total instances as anomalies. (3) Yelp: we regard the reviews of 0 and 1 stars as the positive class, and the reviews of 3 and 4 stars as the negative class.

For tabular datasets, most of them comes from [14], we directly adopt these datasets. Diabetes dataset is sampled from a dataset on Kaggle[1], we increased the difficulty of learning a model by randomly sampling positive and negative examples at different ratios.

**Real-world datasets.** The details of real-world datasets are as follows:

- CIFAR-10N: It is an image classification dataset, which consists of 50000 samples for training and 10000 for testing. We regard the 'Airplane' as positive class and others as negative class. Following [14], we use ResNet18[16] which is pretrained on the ImageNet[5] to extract embedding after the last average pooling layer. We utilize the extracted 512-dimensional embeddings as features.

- Sentiment: It contains 5000 sentences from movie reviews extracted from the website Rotten-Tomatoes.com and whose sentiment was classified as positive or negative. This dataset is the original one in the website[2].

Table 7: AUC for different annotators with class-dependent noise and instance-dependent noise.

| Type | Class-dependent Noise | | | Instance-dependent Noise | | |
|---|---|---|---|---|---|---|
| | Annotator1 | Annotator2 | Annotator3 | Annotator1 | Annotator2 | Annotator3 |
| AgNews | 0.632 | 0.684 | 0.738 | 0.692 | 0.679 | 0.642 |
| 20News | 0.634 | 0.686 | 0.740 | 0.667 | 0.692 | 0.648 |
| IMDb | 0.632 | 0.684 | 0.738 | 0.624 | 0.607 | 0.605 |
| Yelp | 0.632 | 0.684 | 0.738 | 0.658 | 0.625 | 0.617 |
| Amazon | 0.632 | 0.684 | 0.738 | 0.623 | 0.597 | 0.566 |
| Diabetes | 0.627 | 0.681 | 0.735 | 0.672 | 0.641 | 0.639 |
| Backdoor | 0.643 | 0.694 | 0.746 | 0.683 | 0.761 | 0.746 |
| Campaign | 0.628 | 0.681 | 0.735 | 0.587 | 0.576 | 0.663 |
| Waveform | 0.640 | 0.691 | 0.748 | 0.672 | 0.593 | 0.635 |
| Celeba | 0.634 | 0.687 | 0.749 | 0.667 | 0.642 | 0.594 |
| SVHN | 0.632 | 0.684 | 0.739 | 0.693 | 0.679 | 0.642 |
| F-MNIST | 0.632 | 0.684 | 0.738 | 0.795 | 0.780 | 0.774 |
| CIFAR-10 | 0.632 | 0.686 | 0.738 | 0.629 | 0.666 | 0.540 |

---

[1] https://www.kaggle.com/datasets/julnazz/diabetes-health-indicators-dataset
[2] http://fprodrigues.com//mturk-datasets.tar.gz

**Details of labels.** As mentioned in Section 4.1, we use two methods to simulate three annotators. The AUC of these three set of labels is shown in Table 7.

## C Implementation Details of the Algorithm

The implementation procedure of the proposed method CRL is shown in Algorithm 1.

---

**Algorithm 1** Collaborative Refining for Learning from inaccurate labels (CRL).

---

**Input:** Dataset $D = \{\boldsymbol{x}_i, \{\tilde{y}_i^r\}_{r=1}^R\}_{i=1}^N$. The chosen set of epochs $T$. The proportion of selected samples $p$.

**Output:** Submodels $\{f_{\Theta_r}\}_{r=1}^R$. Final submodel $f_{\Theta_{R+1}}$.

1: // Step 1: Gather dataset $D_d$ and $D_a$.
2: Obtain $D_d$ where annotators disagree, i.e. $\exists r_0, r_1 \subseteq \{1, ..., R\}, \tilde{y}_i^{r_0} \neq \tilde{y}_i^{r_1}$.
3: Obtain $D_a$ where annotators agree, i.e., $\forall r_0, r_1 \subseteq \{1, ..., R\}, \tilde{y}_i^{r_0} = \tilde{y}_i^{r_1}$.
4: Initialize $R$ queues $\{\Theta_r^t\}_{t=1}^{|T|}, r \in \{1, ..., R\}$, each with length $|T|$.
5: **for** $e = 1$ **to** *max_epoch* **do**
6:     // Label Refining for samples with Disagreements (LRD)
7:     // Step 2: Generate the refined dataset $D_d^*$.
8:     Obtain instance-wise loss values from $\{f_{\Theta_r}\}_{r=1}^R$ by Eq. (1).
9:     Obtain instance-wise refined label $\tilde{y}^*$ by Eq. (3) and Eq. (4).
10:     Construct the refined dataset $D_d^*$ with refined label $\tilde{y}^*$.
11:     // Robust Union Selection (RUS)
12:     // Step 3: Generate the refined dataset $D_a^*$.
13:     Obtain instance-wise loss values from submodels with every $\Theta$ in $\{\Theta_r^t\}_{r=1, t=1}^{R, |T|}$ by Eq. (1).
14:     Calculate the instance-wise selection criteria by Eq. (10).
15:     Sort the samples in $D_a$ by the criteria and select the smallest $p$ as the refined dataset $D_a^*$.
16:     // Step 4: Utilize $D_d^*$ and $D_a^*$ to train $\Theta_{R+1}$.
17:     Compute loss $\mathcal{L}_{D_d^* \cup D_a^*}$ on $D_d^*$ and $D_a^*$ with $f_{\Theta_{R+1}}$ by Eq. (1).
18:     Update parameters $\Theta_{R+1}$ with loss $\mathcal{L}_{D_d^* \cup D_a^*}$.
19:     // Step 5: Utilize $D$ to train $\{\Theta\}_{r=1}^R$.
20:     Compute loss $\mathcal{L}_D$ on $D$ with submodels $\{f_{\Theta_r}\}_{r=1}^R$ by Eq. (2).
21:     Update parameters $\{\Theta_r\}_{r=1}^R$ with loss $\mathcal{L}_D$.
22:     // Step 6: Update the queues with the latest parameters.
23:     **if** The queues have reached their capacity **then**
24:         Remove the first element of each queue, e.g., $\{\Theta_r^1\}_{r=1}^R$.
25:     **end if**
26:     Save submodels' latest parameters into their respective queues.
27: **end for**

---

## D Related Work

**Learning from a single inaccurate label** is a well-established topic, and numerous approaches have been proposed [11–13, 17, 19, 23–25, 28, 35]. These works can be divided into two main categories. Some works aim to estimate the noise transition matrix [12, 17, 24, 28], which contains the probabilities of clean labels flipping into inaccurate labels. However, the noise transition matrix is hard to estimate accurately. The second approach is sample selection, employing sample selection methods to select possibly clean examples from a mini-batch and train the model with these examples [11, 13, 19, 23, 25, 35]. Many sample selection methods leverage observed patterns, such as the small-loss filtering criterion that considers samples with small losses as clean. The central challenge in these methods is to construct selection criteria.

**Learning from multiple sets of inaccurate labels.** The works can be divided into two main categories. One stream of research focuses on advanced aggregation algorithms that infer the true labels before or during the training stage, others focus on end-to-end learning, simultaneously learning from multiple sets of inaccurate labels.

In aggregation algorithms, the simplest method is Majority Voting [38], where all the annotations are treated equally and labels are aggregated by voting. Some advanced methods go beyond majority voting. Weighted Majority Voting [20] uses a weight vector to model annotators' expertise, Max-margin Majority Voting [32] combines the concept of majority voting with the principles of margin maximization. Enhanced Bayesian Classifier Combination (EBCC) [22] utilizes the Bayesian classifier to infer truth by modeling the correlation between annotators. By viewing the true labels as a latent variable, some methods considering relations among annotators are proposed to infer true labels based on the Expectation-Maximization (EM) algorithm. The seminal works [4] (known as the DS model) assume that every annotator has class-dependent confusion when providing annotations, which is modeled by an annotator-specific noise confusion matrix. The DS model is the basis of many works for aggregating labels from different annotators with the help of EM algorithm [1, 2, 34, 43].

In recent studies, an increasing number of studies focus on end-to-end learning. CrowdLayer[26] trains an end-to-end Deep Neural Network (DNN) with parametric source-specific transition matrices. DoctorNet[10] aims to train DNNs that exploit multiple annotators' information with different softmax output layers. HyperEnsemble [37] trains several individual models for each set of inaccurate labels (i.e., $k$ models for $k$ sets of labels) and combines their outputs by averaging and maximizing, referred to as HE_A and HE_M, respectively. UnionNet[36] learns a transition matrix for all multiple sets of labels together. CoNAL[3] assumes that the annotation noise is attributed to two sources (common noise and individual noise), and combines these two noises by a Bernoulli random variable. SLF[7] proposes to make the weight vectors and the confusion matrices data-dependent and comes up with two regularization methods for the confusion matrix to guide the training process. ADMoE[45] leverages the Mixture of Experts (MoE) architecture to encourage specialized and scalable learning from multiple inaccurate sources. SDM[30] employs a self-cognition module to identify both instance-wise noise and annotator-wise quality and adopts a mutual-denoising module to aggregate these identifications and accordingly refine the model.

