# OpenReview forum: "Collaborative Refining for Learning from Inaccurate Labels"
_NeurIPS.cc/2024/Conference — NeurIPS 2024 poster_

### Official Review · Reviewer_qood · 2024-07-08

**Soundness:** 1
**Presentation:** 1
**Contribution:** 1
**Rating:** 5
**Confidence:** 4

**Summary:**

In common practical scenarios, autonomous annotators are used to create labeled datasets, reducing the dependence on manual labeling, which can be costly and time-consuming. Learning methods leverage multiple weak labels to annotate large amounts of data, though these weak labels are often noisy and imperfect. The paper presents a collaborative refining approach for learning from these inaccurate labels. To refine the data, the authors differentiate between cases where multiple annotators agree or disagree on the labels for a given sample. For samples with disagreements among annotators, the authors propose a noise-filtering method, while for samples with unanimous agreement, they suggest an aggregating method.

**Strengths:**

The paper is well written and easy to follow.

The authors compare their method with many others and use multiple datasets.

**Weaknesses:**

My main issue with this paper is that the problem is not well explained, there is a lack of literature, and most experiments are not conducted with real datasets. If I understand correctly, the problem addressed involves unlabeled samples and the output from multiple annotators. The goal is to obtain labels for these unlabeled samples. This problem is commonly referred to as programmatic weak supervision [1]. In programmatic weak supervision, the objective is to obtain probabilistic labels for the unlabeled samples, and models used to generate these label probabilities are commonly known as label models. Subsequently, these labels are utilized to train a classifier (end model).

In the problem formulation, it is initially stated that the aim is to obtain labels for unlabeled samples, and then (line 85) it is mentioned that the goal is to learn a classifier. What is the objective of the problem being addressed?

If the objective is to obtain labels for unlabeled samples, why aren't probabilistic labels obtained as in programmatic weak supervision?

The WRENCH library [2] collects multiple datasets with outputs from multiple annotators, along with state-of-the-art methods. Why weren't the annotators from that library used?

In the experiments, comparisons are made with methods such as EBCC, IBCC, and WeaSEL. As far as I know, EBCC and IBCC are label models, whereas WeaSEL is a joint model (label model + end model). Therefore, the objectives of these methods are not the same.

References

[1] Zhang, J., Hsieh, C. Y., Yu, Y., Zhang, C., & Ratner, A. (2022). A survey on programmatic weak supervision. arXiv preprint arXiv:2202.05433.

[2] Zhang, J., Yu, Y., Li, Y., Wang, Y., Yang, Y., Yang, M., & Ratner, A. (2021). WRENCH: A comprehensive benchmark for weak supervision. arXiv preprint arXiv:2109.11377.

Ratner, A., Bach, S. H., Ehrenberg, H., Fries, J., Wu, S., & Ré, C. (2017, November). Snorkel: Rapid training data creation with weak supervision. In Proceedings of the VLDB endowment. International conference on very large data bases (Vol. 11, No. 3, p. 269). NIH Public Access.

Ratner, A. J., De Sa, C. M., Wu, S., Selsam, D., & Ré, C. (2016). Data programming: Creating large training sets, quickly. Advances in neural information processing systems, 29.

Mazzetto, A., Cousins, C., Sam, D., Bach, S. H., & Upfal, E. (2021, July). Adversarial multi class learning under weak supervision with performance guarantees. In International Conference on Machine Learning (pp. 7534-7543). PMLR.

Balsubramani, A., & Freund, Y. (2015, June). Optimally combining classifiers using unlabeled data. In Conference on Learning Theory (pp. 211-225). PMLR.

**Questions:**

What happens if an annotator chooses to abstain from labeling a sample?

Is it realistic for all annotators to provide the same label? In the weak supervision datasets such as WRENCH datasets, this rarely occurs.

Building on the previous question, what if one of the datasets $D_d$ or $D_a$ is empty?

What assumptions are made about the annotators?

**Limitations:**

See above

---

> ### Author Rebuttal · Authors · 2024-08-07
>
> Thank you for your feedback. We recognize that there may be some misunderstandings regarding our paper and would like to take this opportunity to clarify these points and address your concerns.
>
> **Main issue 1: "My main issue with this paper is that the problem is not well explained, there is a lack of literature."**
>
> The problem is explicitly stated in our manuscript, specifically in lines 1 and 31-32: "This paper considers the problem of learning from multiple sets of inaccurate labels." Our paper primarily explores training a model using multiple sets of noisy labels to make predictions for unseen samples (the joint model you mentioned). The entire process of our method is described multiple times in the paper, e.g., lines 8-16, 106-108, 319-325.
>
> This is a well-established and clearly-defined research topic with substantial contributions from various researchers [1-10]. A comprehensive summary of these works is provided in Lines 30-47. These works represent widely acknowledged methods of this topic, and we have provided an overview of them as well as conducted a comprehensive comparison in our experiments. The literature and compared algorithms discussed in the most recent works [10] (AAAI2024) and [12] (ICML2024) have been adequately covered in this paper as well.
>
> **Main issue 1: "If I understand correctly, the problem addressed involves unlabeled samples and the output from multiple annotators. The goal is to obtain labels for these unlabeled samples."**
>
> It seems that there might be a misunderstanding regarding our research focus. The problem we are addressing has significant differences from what you describe. While your description aligns with inferring labels for existing samples (label model), our paper primarily explores training a model using multiple sets of noisy labels to make predictions for unseen samples (joint model). This objective is highlighted in several places in our manuscript, including Lines 1, 14-16, 85, and 106-108. and this is also the common focus of many works [1-4, 6-10, 12].
>
> The methods you mentioned, such as EBCC and IBCC, are targeted as one important line of compared works. In our experiments, we utilize generated labels from these label models to train a Multi-Layer Perceptron (MLP) for a fair comparison, which follows previous works [4, 6, 7, 8, 9, 10, 12]. Moreover, our comparison includes a broader range of methods, ensuring a comprehensive evaluation of our proposed approach.
>
> **Main issue 3: "Most experiments are not conducted with real datasets" and about WRENCH.**
>
> Our experiments have included CIFAR10N and Sentiment Polarity, which were two real-world datasets published on Amazon Mechanical Turk for annotation and were utilized in the most recent works like [10] (AAAI 2024) and [12] (ICML 2024). The results are summarized in Table 2 (Page 7), substantiating the reliability and applicability of our approach in handling real-world noisy labels.
>
> When we face the problem of learning from noisy labels, due to the uncontrollable noise levels in real-world datasets, experiments are typically first validated under controlled noise conditions on benchmark datasets, before being further tested on real-world datasets. Many studies follow this approach [1-12], and we conducted our experiments in the same way.
>
> We also use 13 benchmark datasets, some of which are featured in recent studies, such as [9] (AAAI 2023) and [12] (ICML 2024). Hence, we believe our experiments are relatively comprehensive. Since we followed recent works on this topic and our experiments were relatively thorough, we did not use the WRENCH library at that time. We will consider incorporating some WRENCH datasets and adding the results in the revised version.
>
> **Reference**
>
> [1] Aggnet: deep learning from crowds for mitosis detection in breast cancer histology images. TMI 2016.
> [2] Deep learning from crowds. AAAI2018.
> [3] Who said what: Modeling individual labelers improves classification. AAAI2018.
> [4] Max-mig: an information theoretic approach for joint learning from crowds. arxiv 2019.
> [5] Exploiting worker correlation for label aggregation in crowdsourcing. ICML2019.
> [6] Coupled-view deep classifier learning from multiple noisy annotators. AAAI2020.
> [7] Learning from crowds by modeling common confusions. AAAI2021.
> [8] Learning from crowds with mutual correction-based co-training. ICKG 2022.
> [9] Admoe: Anomaly detection with mixture-of-experts from noisy labels. AAAI2023.
> [10] Coupled confusion correction: Learning from crowds with sparse annotations. AAAI2024.
> [11] End-to-end weak supervision. NIPS2021.
> [12] Self-cognitive Denoising in the Presence of Multiple Noisy Label Sources. ICML 2024.
>
> **Question 1: What happens if an annotator chooses to abstain from labeling a sample?**
>
> Under the binary classification scenario, as considered in our paper, annotations can be swiftly generated with no sparsity, e.g., by multiple rules. Even when such sparsity is present, it can often be generally addressed by defaulting to the majority class.
>
> **Question 2: Is it realistic for all annotators to provide the same label? In the weak supervision datasets such as WRENCH datasets, this rarely occurs.**
>
> It is realistic. For instance, CIFAR-10N and Sentiment Polarity datasets are both manually labeled via Amazon Mechanical Turk, in which 58.28% and 60.36% of the samples, respectively, are given the same labels by all the annotators.
>
> **Question 3: Building on the previous question, what if one of the datasets $D_d$ or $D_a$ is empty?**
>
> In this case, we can only use LRD in our framework to deal with $D_d$ or only use RUS to deal with $D_a$.
>
> **Question 4: What assumptions are made about the annotators?**
>
> In this paper, we focus on the labels themselves. The two most important assumptions are (1) class-conditional noise assumption and (2) the diagonal elements of noise transition matrices are greater than 0.5. These assumptions are widely used in many papers.

---

> > ### Comment · Reviewer_qood · 2024-08-12
> >
> > Thank you for answering all my questions. My doubts have been solved. After reading the comments from other reviewers and the reviewers' responses I have decided to raise my score to a 5.

---

> > > ### Author Response · Authors · 2024-08-12
> > >
> > > We are sincerely grateful for your reevaluation of our paper, thanks for your time.

---

### Official Review · Reviewer_qnq5 · 2024-07-10

**Soundness:** 3
**Presentation:** 3
**Contribution:** 2
**Rating:** 5
**Confidence:** 3

**Summary:**

This paper proposes a collaborative refining approach for learning with inaccurate labels provided by low-cost annotators, such as rule-based systems. It introduces strategies based on annotator agreement to filter out noise and enhance data quality. The method includes comparative filtering for conflicting labels and aggregation for consistent annotations, all guided by theoretical bounds on loss values. Extensive experiments on various datasets demonstrate significant improvements in learning performance despite the presence of label inaccuracies.

**Strengths:**

- The paper introduces a collaborative refining method for handling inaccurate labels obtained from low-cost annotators, such as rule-based systems, in contrast to traditional label aggregation approaches.

- This paper proposes strategies based on annotator agreement to filter out inaccuracies through comparative analysis and aggregation, thereby enhancing the quality of the training data.

- Theoretical analysis uncovers relationships among multiple sets of labels, corresponding models, and true labels, providing a foundation for reliable label selection.

- Extensive experiments conducted on various benchmark and real-world datasets demonstrate the effectiveness of the proposed methods in improving learning performance despite label inaccuracies.

**Weaknesses:**

- The proposed method of this paper seems similar to some research methods explored in unreliable partial label learning. Is there any relations between the two?

- When the model is not the Bayesian optimal model, why does $\ell(f_{\theta_0^*}(x),\tilde y^0)<\ell(f_{\theta_1^*}(x),\tilde y^1)$ hold in Theorem 1?

**Questions:**

Please check the weaknesses, and some minor comments are presented below:

- The reference format of the equation is inconsistent, sunch as some are written as Eq. 1, while others are written as Eq. (10).

---

> ### Author Rebuttal · Authors · 2024-08-07
>
> Thank you for your insightful comments and questions. We hope our response can satisfactorily address your questions.
>
> **Weakness 1: The proposed method of this paper seems similar to some research methods explored in unreliable partial label learning. Is there any relations between the two?**
>
> In terms of the overall goal—extracting useful information from imperfect data—our proposed method is consistent with unreliable partial label learning, which is also a common goal for other works in our field. However, our method differs significantly from the methods explored in unreliable partial label learning. Guided by theoretical insights, we train multiple submodels and utilize the relationships among these submodels, multiple sets of noisy labels, and true labels to make label decisions and sample selections. To the best of our knowledge, there is no similar work in unreliable partial label learning that addresses the problem from this angle. This significant difference from previous methods is the key contribution of our method.
>
>
> **Weakness 2: When the model is not the Bayesian optimal model, why does $\ell \( {f_{\Theta_0^\star} }{(x)},\tilde{y}^0 \) < \ell \({f_{\Theta_1^\star} }{(x)},\tilde{y}^1\)$ hold in Theorem 1?**
>
> It is challenging to theoretically prove the inequality holds when the model is not the optimal one. We provide an intuitive explanation and experimental verifications here. The training of the model to its optimal state is a gradual process. As the model progressively approaches optimal state during training, its parameters and structure stabilize, and its behavior and performance tend to converge towards that of the theoretical optimal model. At this point, the effects of Theorem 1 begin to manifest.
>
> We've provided our experimental verifications in Table 4 (page 9). The results show that after training for 100 steps, the label quality produced by LRD (based on Theorem 1) is significantly higher than the original labels (average AUC 0.762 vs 0.368 on $D_d$). With ongoing training, label quality rises (e.g., average AUC 0.819 on $D_d$ at 500 steps).
> This evidence suggests that:
>
> ● Theorem 1 begins to be valid after training for some steps rather than being effective only in the optimal state.
>
> ● The validity of Theorem 1 improves gradually during the convergence process, which is consistent with our intuitive explanation.
>
> These verifications bridge theory and practical application, offering considerable flexibility in choosing when to use the refined labels.
>
> **Question 1: Please check the weaknesses, and some minor comments are presented below: The reference format of the equation is inconsistent, such as some are written as Eq. 1, while others are written as Eq. (10).**
>
> Thanks for your suggestions, we will improve it in the revised version.
>
> Should there be any remaining questions or points of clarification required, we would be more than willing to provide further details or engage in additional discussion.

---

> > ### Comment · Reviewer_qnq5 · 2024-08-12
> >
> > Thank you for thoroughly addressing my questions and clarifying my doubts. After considering the feedback from other reviewers, I have decided to maintain my score.

---

> > > ### Author Response · Authors · 2024-08-12
> > >
> > > We appreciate your feedback, which is valuable for improving the quality of our research. Thank you for the time and effort you have dedicated to reviewing our work.

---

### Official Review · Reviewer_hvQ6 · 2024-07-10

**Soundness:** 3
**Presentation:** 3
**Contribution:** 3
**Rating:** 7
**Confidence:** 3

**Summary:**

This paper considers binary classfication from multiple sets of noisy labels, focusing on data refinement to generate clean labels. At each step of training, It proposes to first separate the dataset by whether label disagreement exists, and then tackle each subset using different methods. For the subset with disagreed labels, authors propose to follow the label with lowest lost. For the subset with all same labels, authors propose a delicated designed term to filter out unreliable data points. Thorough empirical evaluation shows good performance under various settings.

**Strengths:**

- The proposed method is well-motivated by theory and of high practical interest.
- The overall presentation of the paper is well-written and clear enough to understand.
- Comprehensive empirical evaluation has been carefully conducted and the performance of the proposed method has been clearly demonstrated.

**Weaknesses:**

- Limitations such as binary classification and class-conditional noise assumption for LRD can be addressed more clearly in introduction.
- Discussions on model architecture selection can be improved. The design of using R + 1 heads with a shared backbone first appears in section 2 without any detailed introduction.
- Algorightm 1 is mentioned several times in the paper but itself is located in Appendix. This draws concern on abusing the page limit.

**Questions:**

- Is it possible to lower the bar of annotator agreement? In other words, is it possible that labels with few disagreement being filtered out using RUS, so it is not neccessary to keep all labels the same before RUS?
- LRD is designed under class-conditional noise assumption, but empirically performs better for instance-dependent noise. Is there any empirical observation deserves to discuss?
- At the bottom of page 5, it is an interesting observation that relatively large variance are selected. How selection of $\phi(x)$ would change this behavior? What leads to the adoption of this function based on Taylor expansion?

**Limitations:**

Limitations are properly addressed in the paper.

---

> ### Author Rebuttal · Authors · 2024-08-07
>
> Thank you for the time and effort you have dedicated to reviewing our work. We appreciate your insightful comments and questions, which are valuable for improving the quality of our research.
>
> **Weakness 1: Limitations such as binary classification and class-conditional noise assumption for LRD can be addressed more clearly in introduction.**
>
> Thank you for your suggestion. We will discuss these limitations in the introduction in the revised version.
>
> **Weakness 2: Discussions on model architecture selection can be improved. The design of using R + 1 heads with a shared backbone first appears in section 2 without any detailed introduction.**
>
> Due to page limitations, we have provided a brief description of the model structure. We will provide a more detailed explanation in the revised versions. We design such a shared backbone to enhance information sharing and to reduce the overall computational cost of the method. With the shared backbone, each sub-model only requires a simple three-layer MLP.
>
> **Weakness 3: Algorightm 1 is mentioned several times in the paper but itself is located in Appendix. This draws concern on abusing the page limit.**
>
> Thank you for your suggestion, we will improve it in the revised version.
>
> **Question 1: Is it possible to lower the bar of annotator agreement? In other words, is it possible that labels with few disagreement being filtered out using RUS, so it is not neccessary to keep all labels the same before RUS?**
>
> Yes, it's an interesting and sensible idea. For example, we can introduce a new hyperparameter: the consistency rate. For samples where the consistency rate is higher than the threshold, the RUS method is used, and for samples where the consistency rate is lower than the threshold, the LRD method is employed. Since setting this threshold is related to the quality of the dataset itself and the number of label sets, it is more suitable for practical scenarios, especially when we have a good understanding of the label quality.
>
> **Question 2: LRD is designed under class-conditional noise assumption, but empirically performs better for instance-dependent noise. Is there any empirical observation deserves to discuss?**
>
> This is a complex question, and we try to provide some tentative insights here.
>
> On the one hand, class-dependent noise is random and unrelated to features, while instance-dependent labels are related to the features and inherently contain the annotator's intelligence, thus providing more information. This makes instance-level labels more conducive to learning when the proportion of correctly labeled samples is close. This phenomenon aligns with our experiments, where the compared algorithms also tend to perform better under instance-dependent noise conditions. This suggests that the additional information contained in instance-dependent noisy labels can indeed facilitate learning.
>
> On the other hand, our algorithm conceptually integrates the intelligence of multiple models and leverages information from multiple sets of labels. This design enhances the robustness and flexibility of our algorithm, allowing it to adapt to different noise scenarios. Our experiments demonstrate that our algorithm not only performs well on benchmark datasets but also shows competitive performance on real-world datasets.
>
>
>
> **Question 3: At the bottom of page 5, it is an interesting observation that relatively large variance are selected. How selection of 𝜙(𝑥) would change this behavior? What leads to the adoption of this function based on Taylor expansion?**
>
> $\phi(x)$is used for a basic smoothing. The significance of this smoothing is that it reduces the impact of outliers when calculating the mean. Therefore, the current method is a good choice. We speculate that other smoothing methods, such as truncating outliers, could also be effective.

---

> > ### Comment · Reviewer_hvQ6 · 2024-08-11
> > **keeping my score**
> >
> > I have read through other reviewers comments and corresponding authors response. I thank authors for their patient and detailed response.
> > For my concerns, authors kindly considers adjusting the degree of annotator disagreement. For question2, authors disentangled my conception of class-conditional and instance-dependent assumptions.
> > I believe this is a manuscript worth to evaluate, thus I would like to keep my score.

---

> > > ### Author Response · Authors · 2024-08-11
> > >
> > > We would like to extend our heartfelt thanks for the time you spent reviewing our responses to the comments from all reviewers and for your continued support. We highly concur with and appreciate the ideas you suggested to lower the bar of annotator agreement. We will ensure that the final version of the manuscript reflects all the valuable feedback we have received, including yours, to further enhance its quality.

---

### Official Review · Reviewer_eKLA · 2024-07-13

**Soundness:** 3
**Presentation:** 2
**Contribution:** 3
**Rating:** 6
**Confidence:** 3

**Summary:**

This paper introduces a framework for learning from inaccurate labels obtained from multiple annotators. It utilizes annotator agreement to assess label reliability and applies two strategies: one for samples with annotator disagreements (LRD) and another for samples where all annotators agree (RUS). In both cases, the framework uses a number of submodels equal to the number of annotators (with shared layers). The framework refines unreliable datasets into relatively reliable datasets that are used to train the final model. LRD selects reliable labels by comparing the losses of the submodels and choosing the label assigned by the submodel with the smallest loss. RUS enhances dataset quality using a different loss-based selection criterion. Experiments are conducted on class-dependent, instance-dependent, and real-world datasets.

**Strengths:**

1) The analysis conducted is both interesting and original.
2) The paper is clearly written and is easy to follow.

**Weaknesses:**

1) The absence of multiple runs or statistical measures such as standard deviation limits the robustness and reliability of the reported results.

2) Table 1 does not provide information on the quality or reliability of the annotations used in the presented setting.

3) Including additional comparative methods, such as Dawid Skene [1], Iterative Weighted Majority Voting [2], and [3], would enrich the analysis and provide a broader evaluation of the proposed approach.

[1] Dawid, A. P. and Skene, A. M. (1979). Maximum likelihood estimation of observer error-rates using the em algorithm. Journal of the Royal Statistical Society: Series C (Applied Statistics), 28(1):20– 28.
[2] Li, H. and Yu, B. (2014). Error rate bounds and iterative weighted majority voting for crowdsourcing. arXiv preprint arXiv:1411.4086.
[3]  Karger, D. R., Oh, S., and Shah, D. (2014). Budget-optimal task allocation for reliable crowdsourcing systems. Operations Research, 62(1):1–24.

**Questions:**

- What are the advantages of using $\phi$ in equation 7 instead of directly using the loss?

- Has a comparison regarding the running time of the proposed method and the baselines been conducted?

**Limitations:**

Yes

---

> ### Author Rebuttal · Authors · 2024-08-07
>
> We would like to express our sincere gratitude to you for your insightful comments and questions.
>
> **Weakness 1: The absence of multiple runs or statistical measures such as standard deviation limits the robustness and reliability of the reported results.**
>
> Thank you for pointing out the necessity of including standard deviation to reflect the statistical significance of our results. The results reported in our submission currently represent the average outcome of five results. Due to space constraints in the current formatting, we have only presented the mean values. We will include the variance in the final version of our paper.
>
> **Weakness 2: Table 1 does not provide information on the quality or reliability of the annotations used in the presented setting.**
>
> Due to page limits, this information can be found in Table 7 on page 16 in the Appendix.
>
> **Weakness 3: Including additional comparative methods, such as Dawid Skene [1], Iterative Weighted Majority Voting [2], and [3], would enrich the analysis and provide a broader evaluation of the proposed approach.**
>
> Thank you for your suggestions. These methods [1][2][3] are algorithms that have provided some inspiration for the current methods. New approaches have made further advancements based on the ideas of these methods. For instance, the concept of weighting different label sources, as discussed in [2], has been significantly expanded upon in [4]. These newer approaches have been thoroughly compared and evaluated in the paper, therefore, we did not include these algorithms. We will consider adding a comparison to enrich the analysis in the revised version.
>
> [1] Maximum likelihood estimation of observer error-rates using the em algorithm. Journal of the Royal Statistical Society: Series C (Applied Statistics), 1979.
> [2] Error rate bounds and iterative weighted majority voting for crowdsourcing. arXiv 2014.
> [3] Budget-optimal task allocation for reliable crowdsourcing systems. Operations Research, 2014.
> [4] Who said what: Modeling individual labelers improves classification. AAAI 2018.
>
> **Question 1: What are the advantages of using $\phi$ in equation 7 instead of directly using the loss?**
>
> Equation 7 utilizes a smooth function that narrows the range of the loss. Compared to directly using the loss, this approach can initially help mitigate the impact of outliers. Additionally, by introducing a lower bound for the losses, considering variances, we can retain clean samples that might otherwise be excluded due to abnormal predictions by certain submodels.
>
> **Question 2: Has a comparison regarding the running time of the proposed method and the baselines been conducted?**
>
> As you suggested, we ran our method and some competitive baselines 10 times on 5 datasets and recorded the average running time. All the following experiments were conducted on an Apple computer with an M1 chip and 16GB of memory, maintaining the same parameters as in the main experiments, i.e., the same number of epochs, hidden size, and batch size. The results are as follows, the running time of our method does not show a significant difference compared to other competitive baselines.
>
> |Methods|CoNAL|ADMOE|SLF|Ours|
> |---|---|---|---|---|
> |Seconds per step|0.55|0.65|0.84|0.78|
>
>  Moreover, our method offers room for further acceleration. One potential strategy could be to freeze the samples that have been refined through LRD and RUS after a few epochs. Then, these refined samples could be used only to train the final submodel while freezing the other submodels. Additionally, the computational load is indeed related to the number of label sets used. We can consider initially aggregating some labels using methods like majority voting to alleviate the computational burden.

---

> > ### Comment · Reviewer_eKLA · 2024-08-12
> >
> > Thank you for addressing my questions and doubts about the paper. I have reviewed the authors' rebuttal, along with the other reviewers' comments. After careful consideration, I have decided to keep my score.

---

### Official Review · Reviewer_9hER · 2024-07-14

**Soundness:** 3
**Presentation:** 3
**Contribution:** 2
**Rating:** 5
**Confidence:** 4

**Summary:**

This paper studied learning from multiple noisy labels via data refinement. It first uses the annotator agreement as an instrument to divide all samples into the samples where some annotators disagree and the samples where all annotators agree. Then, a comparative strategy is proposed to filter noise in the samples where some annotators disagree, and a robust union selection is used to select clean samples in the samples where all annotators agree. Experiments on multiple datasets show the effectiveness of the proposed method.

**Strengths:**

1. The data refinement is important in this age of big data.
2. The label refining for samples with disagreements is novel in learning with noisy labels with multiple annotators.
3. The proposed method can cooperate with other learning-from-crowds algorithms.

**Weaknesses:**

1. The novelty of robust union selection is limited, since it is very similar to robust mean estimation in [1]. Could the authors clarify this?

2. As mentioned in this work, some methods have adopted the small-loss criterion to refine data [2-4]. The comparison between the proposed method and one of them is necessary, which can show the necessity of designing the new data refinement way with multiple annotators. Besides, one latest work [5] in ICML 2024 also explored this direction, I think this work should at least discuss the differences with it.

3. The proposed method seems also suitable for multi-class classification problems. Since many real-world cases are multi-class classification problems, why does this work only focus on binary classification problems?

[1] Sample Selection with Uncertainty of Losses for Learning with Noisy Labels. ICLR 2022

[2] Coupled-view deep classifier learning from multiple noisy annotators. AAAI 2020

[3] Learning from crowds with mutual correction-based co-training. ICKG 2022

[4] Coupled confusion correction: Learning from crowds with sparse annotations. AAAI 2024

[5] Self-cognitive Denoising in the Presence of Multiple Noisy Label Sources. ICML 2024

**Questions:**

1. How does the proposed method cooperate with other algorithms in Table 5?
2. Why the robust union selection is called an aggregating strategy in Line 12 and Line 62?

**Limitations:**

1. The proposed methods need to train R + 1 submodels, which need a large amount of computing resources when R is large.
2. The proposed methods may be not adapted to the annotation sparse case, which is common in learning from crowds.

---

> ### Author Rebuttal · Authors · 2024-08-07
>
> We appreciate your insightful feedback, which is valuable for improving the quality of our research. We hope our response can satisfactorily address your questions.
>
> **Weakness 1: The novelty of robust union selection is limited, since it is very similar to robust mean estimation in [1]. Could the authors clarify this?**
>
> Our basic idea is consistent with [1], which is to use multiple sets of predictions to identify more reliable samples. [1] focuses on scenarios with single noisy labels, utilizing the robust mean of predictions made by the model at different training stages to filter samples. While in the field of learning from multiple sets of labels, to the best of our knowledge, there have been no relevant attempts. Aiming at the characteristics of this problem, our contribution is to collaboratively leverage multiple submodels' abilities to identify more reliable samples and consider the potential issue of abnormal predictions from some submodels.
>
> Specifically, robust union selection not only incorporates robust mean estimation, using it as a very fundamental smoothing step, but also goes further by introducing the variance of losses from different submodels to form a selection criterion. This criterion allows clean samples that might have been overlooked due to higher mean losses resulting from inaccurate predictions of certain submodels to potentially be re-included in the training process, rather than being excluded. In contrast, [1] follows the smoothing step by introducing the number of times each sample has been used in training, ensuring that less utilized samples are given more opportunities to be included. These are different solutions aimed at different problems.
>
> **Weakness 2: As mentioned in this work, some methods have adopted the small-loss criterion to refine data [2-4]. The comparison between the proposed method and one of them is necessary, which can show the necessity of designing the new data refinement way with multiple annotators. Besides, one latest work [5] in ICML 2024 also explored this direction, I think this work should at least discuss the differences with it.**
>
> Following your suggestions, we have incorporated a comparison with [2], abbreviated as CVL. We present average AUC results under class-dependent noise here:
> |Methods|Agnews|20News|Yelp|IMDb|Amazon|Diabetes|Backdoor|Campaign|Waveform|Celeba|SVNH|FMNIST|CIFAR10|
> |-|-|-|-|-|-|-|-|-|-|-|-|-|-|
> |CVL|0.837|0.828|0.817|0.748|0.695|0.705|0.755|0.752|**0.868**|0.868|0.743|0.752|0.638|
> |Ours|**0.855**|**0.849**|**0.867**|**0.766**|**0.775**|**0.728**|**0.937**|**0.783**|0.840|**0.891**|**0.761**|**0.776**|**0.655**|
>
> [5] is publicly released at ICML 2024 in July, which is later than the NIPS 2024 submission deadline. Following your suggestions, we try to discuss the differences with it.
>
> [5] treats all samples in the same way without distinction. In contrast, our work differentiates samples based on annotator agreement and designs relatively appropriate methods for each. Specifically, for samples where some annotators disagree, we directly determine the relatively accurate label based on theoretical considerations. For samples where all annotators agree, we robustly integrate multiple model predictions to select relatively reliable samples. These are two distinct perspectives. In the revised version of our paper, we will consider adding a comparison with [5].
>
> **Weakness 3: The proposed method seems also suitable for multi-class classification problems. Since many real-world cases are multi-class classification problems, why does this work only focus on binary classification problems?**
>
> In theory, our method is suitable for multi-class classification problems. However, we've encountered some difficulties during practical experiments. The primary issue arises with samples where annotators disagree, particularly when all the given labels for a particular sample are incorrect. Our current method, LRD (Label Refining for samples with Disagreements), struggles to address this situation. It tends to select one of these incorrect labels as the inferred label, negatively impacting the model's performance. In binary classification problems, this situation does not occur because when annotators disagree, one of the labels must be correct.
>
> We are still working on modifying our approach to make it suitable for multi-class classification problems. Based on our framework, A possible approach could be to develop a more refined measure based on annotator agreement, which could be utilized to decide whether to discard these samples, use LRD, or use RUS (Robust Union Selection). Since these ideas are not yet mature enough, we did not include this part in the current work. This will be the focus of our future exploration.
>
> **Question 1: How does the proposed method cooperate with other algorithms in Table 5?**
>
> Thank you for your attention to an important characteristic of our algorithm: its ability to cooperate with other algorithms. After training our model for several epochs, through LRD and RUS, we can produce a refined dataset. We extract the refined dataset and complement it with the original noisy labels. Then the refined dataset containing both the original noisy labels and the refined labels can be used to train other methods in the same manner as in the main experiments. This process is detailed in lines 309-310 of our paper.
>
> **Question 2: Why the robust union selection is called an aggregating strategy in Line 12 and Line 62?**
>
> The "aggregating strategy" mentioned here refers to the concept of harnessing the collective intelligence of multiple submodels to select samples. Specifically, we aggregate the predictions of multiple submodels by calculating the robust mean and the variance, and use our selection criteria to filter samples.

---

> ### Comment · Reviewer_9hER · 2024-08-12
>
> Thanks for the response of the authors. It addressed my major concerns. I decide to increase my score to 5.
> Besides, there are some minor questions and suggestions:
> - How to choose the hyperparameter p in robust union selection?
> - It seems that $f_{\Theta_k^*}$ has not been defined clearly.
> - When cooperating with other algorithms, is the refined data are regards as a new annotators? Does the new formed dataset have the same size with the original dataset?
> - It is better to unify the terms of the proposed strategies or make them clearer in the whole paper. In Abstract and Introduction, a comparative strategy and an aggregating strategy are mentioned, while they don't appear and are not explained in the Method
>  section.
> - To make the compared baselines clearer, I suggest to rename baselines that are training with the results of the label model (e.g.  Majority Voting, EBCC), with the type of end model (classifier). For example, If Majority Voting, EBCC baselines are training deep neural networks with the results from  Majority Voting, EBCC, it is better to name them by NN-MV, NN-EBCC as [6], or by DL-MV, DL-EBCC as [7],  these names can make the readers of the whole community easily understand the focused problem.
>
> [6] Deep learning from crowdsourced labels: Coupled cross-entropy minimization, identifiability, and regularization. ICLR 2023
>
> [7] Transferring annotator- and instance-dependent transition matrix for learning from crowds. TPAMI 2024

---

> ### Author Response · Authors · 2024-08-13
>
> Thank you again for your insightful and constructive suggestions, especially regarding renaming the baselines and adding comparative baselines related to the small-loss criterion, which are of great significance for perfecting our work.
>
> **Q1: How to choose the hyperparameter p in robust union selection**
>
> In our experiments, we set this parameter to 0.8 without adjusting for a fair comparison. If you want to choose an optimal hyperparameter p for a specific dataset, it may be necessary to use cross-validation methods to assess the impact of different p values on model performance. Typically, this optimal value varies across different datasets.
>
> Empirically speaking, the selection of p is related to the size of the dataset, and the quality of the labels. We recommend choosing p between 0.5 and 0.8. If the label quality of the dataset is high, it can be further increased to 0.9.
>
> __Q2: It seems that $f\_{\Theta\_k\^\*}$ has not been defined clearly.__
>
> As is defined in the Preliminaries, $f_{\Theta^{\*}}$ is the optimal classifier and $f_{\Theta_k}$ is the model trained with $k$-th label set. $f_{\Theta_k^\*}$ is the optimal model trained with the $k$-th label set.
>
> **Q3: When cooperating with other algorithms, is the refined data are regards as a new annotators? Does the new formed dataset have the same size with the original dataset?**
>
> Yes, we treat the refined labels as contributions from a new annotator. And the size of the newly formed dataset is a littile smaller than the original, as RUS has filtered out some samples.
>
> **It is better to unify the terms of the proposed strategies or make them clearer in the whole paper. In Abstract and Introduction, a comparative strategy and an aggregating strategy are mentioned, while they don't appear and are not explained in the Method.**
>
> Thank you for your detailed suggestion. We will add a description of these two concepts in the Method to make it clearer.
>
> **To make the compared baselines clearer, I suggest to rename baselines that are training with the results of the label model (e.g. Majority Voting, EBCC), with the type of end model (classifier). For example, If Majority Voting, EBCC baselines are training deep neural networks with the results from Majority Voting, EBCC, it is better to name them as NN-MV, NN-EBCC [6], or DL-MV, DL-EBCC [7], these names can make the readers of the whole community easily understand the focused problem.**
>
> Thank you for your thoughtful suggestion. We are indeed planning to make that change, and in the revised versions, we will ensure that the name of the baselines is updated to the clearer version you suggested.

---

### Author Response · Authors · 2024-08-12

Dear Reviewers,

We extend our deepest appreciation for the valuable insights and constructive suggestions you've provided for our submission to NeurIPS 2024. We are glad that reviewers appreciate the novelty and high practical interest of our work, the interesting and original analysis we have done, and the importance of data refinement that we focus on.

Following your suggestions, we have dedicated substantial effort to address each point raised during the review process, including additional experiments as suggested. As the author-reviewer discussion phase is crucial for refining submissions, we kindly encourage you to review our rebuttal responses at your earliest convenience. Any further discussions or suggestions would be greatly appreciated.

We hope that our efforts in addressing the concerns have resonated well with the paper's objectives and have potentially influenced your evaluation. Thank you once again for your dedication and time. We look forward to your valuable feedback.

Warm regards,

Authors of submission 16683 at NeurIPS 2024

---

### Decision · Program_Chairs · 2024-09-25

**Decision:**

Accept (poster)

**Comment:**

This paper proposes a method for learning from multiple sets of inaccurate labels based on collaborative refinement approach. The proposed approach is interesting and novel in learning with inaccurate labels with multiple annotators. The effectiveness of the proposed method is well demonstrated in the experiments. This paper is well written. More discussion on the existing work is needed. I encourage the authors to improve their paper by reflecting the reviewers’ comments and the author response.